# Non-Invasive Vaccines: Challenges in Formulation and Vaccine Adjuvants

**DOI:** 10.3390/pharmaceutics15082114

**Published:** 2023-08-09

**Authors:** Sumin Han, Panjae Lee, Hyo-Jick Choi

**Affiliations:** Department of Chemical and Materials Engineering, University of Alberta, Edmonton, AB T6G 1H9, Canada; sumin3@ualberta.ca (S.H.); panjae@ualberta.ca (P.L.)

**Keywords:** non-invasive, vaccines, oral, intranasal, transcutaneous

## Abstract

Given the limitations of conventional invasive vaccines, such as the requirement for a cold chain system and trained personnel, needle-based injuries, and limited immunogenicity, non-invasive vaccines have gained significant attention. Although numerous approaches for formulating and administrating non-invasive vaccines have emerged, each of them faces its own challenges associated with vaccine bioavailability, toxicity, and other issues. To overcome such limitations, researchers have created novel supplementary materials and delivery systems. The goal of this review article is to provide vaccine formulation researchers with the most up-to-date information on vaccine formulation and the immunological mechanisms available, to identify the technical challenges associated with the commercialization of non-invasive vaccines, and to guide future research and development efforts.

## 1. Introduction

Vaccine administration methods can be categorized into invasive, minimally invasive, and non-invasive administration, and conventional vaccine administration primarily relies on invasive administration (e.g., intramuscular, subcutaneous, and intradermal) to maximize vaccine bioavailability through site-specific administration. However, such methods of administration have inherent limitations that compromise their utility (Figure 1). Firstly, liquid formulations used in these methods necessitate a cold chain system, making them susceptible to accidental freezing [1,2]. Certain temperature ranges can drastically reduce the shelf life of vaccines, with examples including the Pfizer-BioNTech (<6 months at −60 to −90 °C and <2 h at 8–30 °C), ModernaTX (<30 days at 2–8 °C and <12 h at 8–25 °C), and Oxford-AstraZeneca (<6 months at 2–8 °C and <2 h at 8–25 °C) SARS-CoV-2 vaccines [3]. mRNA vaccines are especially easily affected by changes in liquid formulation (i.e., pH, buffers, temperature, amount of oxygen, etc.), which can lead to the degradation of the vaccines through hydrolysis [4].

Second, the use of needle-based delivery of liquid vaccines also presents technical and safety concerns. The variability in adipose tissue thickness among individuals necessitates adjusting the needle length based on body weight and sex. For instance, a needle length of 1 inch is recommended for individuals weighing less than 70 kg, 1–1.25 inches for men weighing 70–118 kg and women weighing 70–90 kg, and 1.5 inches for men weighing over 118 kg and women weighing over 90 kg [2,5]. Achieving accurate administration requires specialized personnel and proper setups, posing challenges for vaccination in isolated communities and developing countries with limited resources and personnel. Furthermore, needle-based administration carries the risk of needlestick injuries and the transmission of bloodborne pathogens to the injection site (e.g., hepatitis B, hepatitis C, and human immunodeficiency virus (HIV)) [6].

Third, invasive vaccines primarily elicit systemic immune responses in the lower respiratory tract, unlike natural infections that stimulate robust, both systemic and mucosal immunities [7,8]. While systemic immune responses can provide disease-attenuating or disease-preventing immunity through the production of immunoglobulin G (IgG) [7], the limited mucosal immunity in the lower respiratory tract can hinder the elimination of pathogens in the early stages, potentially exacerbating disease severity. On the other hand, minimally invasive and non-invasive vaccines can achieve mucosal vaccination that initiates both robust systemic and mucosal immunities, thereby efficiently preventing pathogens from developing severe disease [9,10].

Invasive vaccines have demonstrated high stability and safety when administered correctly, leading to their predominant use in present-day vaccine delivery. However, due to the aforementioned limitations, there is a growing demand for the development of alternative vaccines, and minimally invasive and non-invasive vaccines have been actively researched as promising candidates. The scope of this manuscript primarily revolves around non-invasive vaccines, which have undergone more comprehensive studies and research compared to minimally invasive vaccines. This review aims to shed light on the advantages and challenges of non-invasive vaccine approaches, along with conventional invasive counterparts, while providing valuable insights into their formulation strategies.

## 2. Types of Vaccines and Their Formulation

Since Louis Pasteur’s groundbreaking conceptualization of vaccination, significant advancements have been made in the development of vaccines in terms of their stability and effectiveness [11]. To date, seven types of vaccines have been developed: live attenuated, inactivated, replicating viral vector, non-replicating viral vector, DNA, RNA, and subunit vaccines. Each vaccine type leverages distinct immunological mechanisms to trigger adaptive immunity upon administration [12]. Replicable vaccines, including live attenuated vaccines, replicating viral vector vaccines, and replicating nucleic acid vaccines, enter host cells and replicate virulent molecules, which are subsequently taken up by dendritic cells (DCs). In contrast, non-replicable vaccines, such as non-replicating viral vector vaccines, inactivated vaccines, subunit vaccines, and non-replicating nucleic acid vaccines, are directly phagocytosed by DCs [12,13]. Replicable vaccines are characterized by prolonged immunogenicity compared to non-replicable vaccines due to their ability to replicate. Whole-particle-based vaccines, such as live attenuated and inactivated vaccines, exhibit the highest immunogenicity compared to other types of vaccines due to the presence of immunostimulant molecules within the particles, effectively inducing immunity similar to that of natural infections [14,15,16,17,18].

Following the phagocytosis, DCs carrying vaccine particles migrate to the draining lymph node, where they present peptides derived from the vaccine antigens on the major histocompatibility complex (MHC) class II or MHC class I molecules. MHC II activation stimulates CD4^+^ T cells (T helper cells), including T helper (Th) 1 cells and Th2 cells [19]. Th1 cells play a role in cell-mediated immunity by secreting interferon-gamma (IFN-γ), a proinflammatory cytokine that eliminates external pathogenic particles [20,21]. Th1 cells also activate CD8^+^ T cells (cytotoxic T cells) by interacting with MHC I [22]. This interaction leads to the production of granzymes, which induce apoptosis, and the pore-forming protein perforin, which physically damages the target cell membrane, allowing granzymes to enter the cell [23]. In contrast, Th2 cells initiate humoral immunity by producing interleukin-4 (IL-4), IL-5, and IL-13, which promote the proliferation and maturation of B cells. Additionally, Th2 cells produce anti-inflammatory IL-10, which modulates the inflammatory responses of Th1 cells (Figure 2) [20,21].

### 2.1. Live Attenuated Vaccines

Live attenuated vaccines (LAVs) consist of live bacteria or virus particles that have been weakened but retain the ability to initiate immune responses. The production of live attenuated vaccines involves either natural or engineered mutations. In natural production, bacteria or viruses are subjected to multiple passages through different species in a process known as serial passage exposure experiments (SPEs) [24,25]. For example, weakened *Theileria annulata* loses its virulence against its original host, cattle, after undergoing 50–100 passages of in vitro culturing [26]. This loss of virulence is attributed to alterations or selections in gene expression triggered by environmental changes. In the case of viruses, genetic modifications can occur even more rapidly. For example, nucleopolyhedrosis viruses (NPVs) have been shown to lose virulence against three out of their six original hosts after eight passages in *Pseudoplusia includens* [27].

Engineered mutation involves the use of transposon mutagenesis, which facilitates the translocation of repetitive DNA sequences to the host cell nucleus, thereby introducing random mutations in the host gene [28]. Another method, gamma rays, is commonly employed to induce DNA strand breaks and oxidative stress in viruses, leading to single base substitutions. However, such mutagenesis techniques often raise safety concerns, including the potential re-emergence of virulence upon repeated exposures to the same host or the restoration of mutated gene sites [24]. Historical examples of virulence re-emergence include the poliovirus vaccine in 1964, the yellow fever vaccine in 2008, and the rotavirus vaccine in 2009 [29]. As a result, research has been focused on developing methods to prevent pathogens from regaining virulence.

One strategy to minimize the re-emergence of virulence is to selectively choose variants with high genetic fidelity for vaccine development [30]. Another approach is multi-region replacement, often combined with gene targeting methods, which involves increasing the number of mutated regions to decrease the probability of mutations reverting to their original state. For example, codon deoptimization replaces conventional codons with synonymous codons, resulting in the same level of immune responses with reduced virulence and re-emergence [30]. Studies have shown that after achieving 97% codon replacement in the contiguous capsid region of polioviruses, the transmission rate in host cells decreased by more than 10-fold [31]. Undesirable mutations can be minimized using small interfering RNA (siRNA) and microRNA (miRNA), which target specific genes for degradation and repress the translation of respective proteins. Zinc finger (ZF) domains, which are protein motifs with finger-like protrusions that bind to DNA, can also be utilized to target specific genetic sites. ZF domains are typically engineered to express multiple types of domains, such as restriction enzymes, to suppress gene transcription or replication origin-competing domains to reduce virus replication [30].

### 2.2. Inactivated Vaccines

Inactivated vaccines offer a high level of safety compared to live attenuated vaccines, as they are free of concerns regarding virulence reversion and transmission [29]. However, these vaccines tend to have relatively low immunogenicity compared to live attenuated vaccines [29]. Therefore, the design of inactivated vaccines must consider two key factors: complete pathogen inactivation and epitope conservation. Meeting the aforementioned requirements requires a comprehensive study of various virus behaviors, including aggregation, protein crosslinking, protein denaturation, and degradation [32].

Delrue (2012) has described the mechanism and chemical reactions involved in different inactivation treatments, such as formaldehyde, glutaraldehyde, 2,2′-dethiodipyridine, β-propiolactone (BPL), binary ethylene imine, pH, temperature, gamma irradiation, and ultraviolet light [32]. Among these treatments, formaldehyde and β-propiolactone (BPL) have been most commonly used in vaccine development for many years [29]. Formaldehyde primarily acts by deforming the adenine residues of genes and amino acids through monohydroxymethylation (NH-CH_2_OH), which creates a methylene spacer (-CH_2_-) that bridges two independent molecules. On the other hand, BPL primarily alkylates the guanine residues of genes, thereby disrupting the reading of sequences [32]. Unlike formaldehyde, BPL treatments preserve the intact epitopes of antigens, leading to greater immunogenicity [29]. Currently, three inactivated vaccines for SARS-CoV-2 have received FDA approval [33]. All three vaccines (Coronavac—Sinovac, Covaxin (BBV152)—Bharat Biotech International, and COVID-19 vaccine (Vero cell)—Sinopharm) employ BPL for the inactivation of the whole SARS-CoV-2 virus [34,35,36]. However, inactivated vaccines face limitations, as they may exhibit poor clearance of unnecessary viral molecules from the human body, which could interact with and harm host cells. Additionally, the process of culturing whole viruses is considered time-consuming compared to gene-based vaccines that can be rapidly amplified [19]. Consequently, there has been a significant shift in the industry towards simpler vaccine forms, such as recombinant DNA and protein subunits [37].

### 2.3. Replicating and Non-Replicating Viral Vector Vaccines

Replicating and non-replicating viral vector vaccines have been developed to minimize the insertion of pathogenic molecules into the human body. These vaccines utilize different types of vectors to deliver the desired viral genes. Replicating viral vector vaccines employ various vectors such as adenovirus, measles virus, poxviruses, and vesicular stomatitis virus, while non-replicating viral vector vaccines use adenovirus, adeno-associated virus, alphavirus, herpesvirus, and poxviruses [17]. The selection of these vectors is based on their genetic safety, high immunogenicity, and low likelihood of pre-existing immunity in hosts [38].

The main distinction between replicating and non-replicating viral vectors lies in their ability to replicate genes within hosts, and non-replicating viral vectors are engineered to lack gene-replicating sites [39]. As an example, non-replicating adenovirus vectors (such as adenovirus serotype 5, 26, 35) have undergone modifications where the replicating site, Early region 1 (E1), is removed and replaced with an expression cassette that contains a highly active promoter responsible for expressing the inserted foreign gene [40,41]. Similarly, vesicular stomatitis virus (VSV) vectors lack the glycoprotein (G) responsible for attaching to host cells, rendering them unable to infect host cells [38]. Integrase-defective lentiviral vectors (IDLVs) serve as another example of non-replicating viral vectors that lack genes inducing pathogenicity, including replicating genes such as Tat, Rev, Nef, Vif, Vpr, Vpu, Vpx, dUTPase, and open reading frames (ORFs) [42]. However, non-replicating viral vectors require relatively high doses or frequent revaccination due to their inability to replicate [38]. Furthermore, these vectors have the potential to induce mutagenesis in wild-type viruses [43]. In contrast, replicating viral vectors can generate long-lasting immunity even with smaller doses due to their replicability. The replicating viral vectors elicit immune responses similar to natural infections, triggering the production of cytokines and other stimulatory factors that can act as adjuvants [38]. As a result, extensive research has been dedicated to developing replicating viral vector vaccines against diverse viruses (e.g., influenza, HIV, hepatitis B virus) [44]. However, replicating viral vector vaccines raise immune-related safety concerns, particularly in immunocompromised patients. Additionally, the presence of gene-replicating sites in these vectors limits their capacity to accommodate the insertion of small-sized genes (e.g., 7–9 kb for non-replicating adenovirus vectors and 3–4 kb for replicating adenovirus vectors) [17].

### 2.4. DNA Vaccines

The production of DNA vaccines mostly involves genetic modifications of plasmids derived from *Escherichia coli*. Subsequently, the modified plasmids are cultivated under fermentation conditions in an anaerobic condition using glycerol, yeast extract, and MgSO_4_ as carbon, nitrogen, and trace metal sources, respectively. The replicated plasmids can be used directly as genetic vectors after undergoing lysis and purification [45,46]. Conventionally, non-live and non-replicating plasmids have been considered prototypical platforms for DNA vaccines; however, due to their low immunogenicity, replicating DNA vaccines have been considered a potential alternative. In general, replicating DNA vaccines are manufactured by combining eukaryotic promoters (e.g., human cytomegalovirus (CMV) immediate promoters and enhancers) and replicon sequences of alphaviruses [14,47,48]. Furthermore, incorporating additional gene inserts, such as adjuvant coding sequences, has been attempted to enhance the immunogenicity of DNA vaccines [49]. For example, incorporating unmethylated CpG motifs into a DNA vaccine has been shown to function as a type of pathogen-associated molecular pattern (PAMP), which can be utilized to boost vaccine-initiated immune responses [50].

Currently, two types of DNA vaccines have been developed: circular vaccines (i.e., minicircle, minivector, miniknot, etc.) and linear vaccines, including immunological-defined gene expression (MIDGE), Micro-Linear vector (MiLV), and Doggybone^TM^ DNA, which are covalently closed (Figure 3). In the case of minicircle (mc) DNA, recombinases are used to separate a plasmid into an mc DNA molecule (containing the genes of interest) and a miniplasmid (containing the bacterial plasmid backbone). This is followed by degradation of the miniplasmid using restriction enzymes [45,51]. Studies have shown that mc DNA exhibits a significantly higher transfection rate (68%) compared to the parental plasmid (34%) due to its smaller size (3.8 kbp) in contrast to the parental plasmid (8.2 kbp) [52]. Minivector DNA is a single-stranded and supercoiled structure of less than 2 kbp in size and less than 4 nm in diameter, and is produced through treatments with recombinases (site-specific cleavage) and topoisomerase IV (separation of the daughter DNA) [53,54]. The size difference between the miniplasmid and minivector contributes to the ease of the separation process, contributing to the high purity of the minivector DNA [53]. A miniknot is an mc DNA strand treated with topoisomerase II, which enhances its physical strength to prevent linearization during processing. However, the use of miniknots is still in a hypothetical stage.

The MIDGE (2.9 kbp) and MiLV (1.7 kb) vaccines are produced in a similar manner, but they utilize different parental plasmids. This process entails the cleavage of sites containing the genes of interest using a restriction enzyme, followed by the ligation of the sticky ends of the cleaved sites with hairpin oligodeoxynucleotides to stabilize the structure [43,55,56]. The Doggybone^TM^ (2.6 kbp) vaccine is also created using plasmids. In this process, plasmids are initially denatured by NaOH, resulting in the production of single-stranded DNA that acts as a template for rolling circle amplification. The amplified single-stranded DNA is then polymerized to form a double-stranded DNA concatemer. This concatemer is subsequently treated with Te1N protelomerase, which recognizes the telomeric ends (Tel-L and Tel-R) and cleaves the linear DNA while covalently closing it. This step removes the bacterial plasmid backbone from the sites containing the genes of interest [57,58]. However, the yield of mc DNA (0.1–8.84 mg/L) is generally much lower compared to conventional plasmids (2.2 g/L of fermentation), and other engineered plasmids are also expected to have lower yields due to physical loss and size reduction during processing [51]. Additionally, although DNA vaccines have exhibited satisfactory immunogenicity in small animals such as rodents, there is a need for improvement when it comes to large animals and humans [59].

### 2.5. mRNA Vaccine

During the initial stages of nucleic acid-based vaccine development, DNA was considered a preferable option to RNA due to its stability and potential for mass production. However, DNA vaccines have shown limited potency in humans and potential risks of integration of inserted DNA into the host genome [15]. In contrast, mRNA vaccines provide certain advantages over DNA vaccines as there is no concern about integration into the host genome. Additionally, since mRNA vaccines are not required to cross the nuclear membrane to initiate the translation process, they tend to have higher immunogenicity than DNA vaccines [60]. Consequently, numerous trials have been conducted to develop mRNA vaccines, encompassing both non-replicating and replicating mRNA variants.

Conventionally, to achieve mass production of mRNA vaccines, mRNA is initially extracted from pathogen particles and converted into complementary DNA (cDNA) through a process called reverse transcription. The cDNA is subsequently inserted into a plasmid and linearized using restriction enzymes to create a 3′ end sequence to achieve similar structure to the original mRNA, followed by transcription of the cDNA in host cells [61]. Nowadays, in vitro transcription is also widely used in mRNA mass production, with a simpler procedure that consists of the insertion of the target gene into DNA templates followed by RNA amplification [62]. The transcribed mRNA contains specific components. These components include a 5′ cap, which regulates translation along with the poly(A) tail, a 5′ untranslated region that regulates mRNA translation, an open reading frame that represents the gene of interest, a 3′ untranslated region that regulates mRNA translation, and a poly(A) tail that enhances the stability of the mRNA [63,64,65].

Replicating RNA vaccines are primarily developed from alphaviruses. For this, the nsP1-4 genes from alphaviruses are extracted and incorporated into conventional non-replicating RNA. Once inside the host cells, these genes are translated into nsP1-4 proteins, which then assemble into an RNA-dependent RNA polymerase (RdRp) complex. This RdRp complex initiates the replication process for RNA, allowing for the amplification of the target gene [66,67].

In terms of immunogenicity, replicating mRNA vaccines are preferred over non-replicating mRNA vaccines. This preference arises from the fact that replicating mRNA vaccines can replicate and maintain replication sites, thereby potentially enhancing their immunogenicity [66].

mRNA vaccines necessitate the use of delivery vehicles due to their electronegativity and large molecular weight (10^5^–10^6^ Da). The negative charge of mRNA, stemming from its phosphate residues, causes repulsion with the anionic phospholipids in the lipid bilayer membrane of host cells [68,69]. Furthermore, mRNA is three to four times larger than other molecules that can easily enter host cells, such as small interfering RNAs (14 kDa) and antisense oligonucleotides (4–10 kDa) [70].

To address this requirement, lipid nanoparticles and polymeric nanoparticles have been extensively investigated as potential delivery vehicles for mRNA. Specifically, cationic lipids have commonly been used for lipid nanoparticles due to their capacity to readily interact with negatively charged mRNA. However, their use has been linked to liver damage and inflammation resulting from interactions with molecules in the body [68]. To tackle this issue, ionizable lipids have been developed. These lipids carry a positive charge under acidic conditions but become neutral at physiological pH [71]. This ionizable property allows them to associate with mRNA under acidic conditions while reducing toxicity in host cells at neutral pH. To date, five different types of ionizable lipids have been developed: unsaturated ionizable lipids, multi-tail ionizable lipids, ionizable polymer–lipids, biodegradable ionizable lipids, and branch-tail ionizable lipids [71].

Lipid nanoparticles are typically formulated with various types of lipids, including cholesterol, polyethylene glycol (PEG) lipid, and phospholipids, in varying composition ratios. Each component has a distinct purpose. For instance, cholesterol enhances the integrity and rigidity of the particles, while PEG lipid reduces particle aggregation and enhances interaction with ligands on target cells [72]. The properties of phospholipids are determined by lipid saturation. For example, 1,2-distearoyl-sn-glycero-3-phosphocholine (DSPC), which consists of saturated lipids, forms a stable lamellar shape that maintains excellent stability over time. In contrast, unsaturated lipids like 1,2-dioleoyl-sn-glycero-3-phosphocholine (DOPC) and dioleoylphosphatidylcholine (DOPE) undergo a transition from a lamellar shape to an inverted conical shape at room temperature due to their lower phase transition temperature, enhancing the efficiency of cargo release from the vehicle [73].

In addition to lipids, polymeric materials have undergone extensive research for efficient delivery of nucleic acids. Cationic polymers such as poly(lysine), poly(ethylene imine) (PEI), and poly(amidoamine) dendrimers exhibit high transfection efficiency but can be cytotoxic as lipid nanoparticles. Therefore, degradable materials like water-soluble lipopolymer (WSLP) are often combined with cationic polymers to reduce toxicity and prevent accumulation [74,75]. Chitosan, known for its biocompatibility, biodegradability, and low cost, is another promising candidate for mRNA delivery. However, its efficacy is not as pronounced as PEI due to its insolubility at physiological pH [76]. To achieve high transfection efficiency and low toxicity, biodegradable polyesters containing cationic side chains have been synthesized, including poly(L-lactide-co-L-lysine), poly(serine ester), poly(4-hydroxy-L-proline ester), and poly [R-(4-aminobutyl)-L-glycolic acid] [74].

### 2.6. Subunit Vaccine

Subunit vaccines have been developed to address safety concerns related to whole-particle-based vaccines. These vaccines can be categorized as protein, polysaccharide, conjugate, virus-like particle (VLP), and toxoid vaccines [29,77]. The production of protein-based subunit vaccines involves the harvest and purification of antigens from bacteria using recombinant DNA encoding the target protein. This is often followed by coupling with carriers (i.e., liposome, polymeric nanoparticles, etc.) to enhance the antigen’s stability against environmental changes in the body, such as pH variations and enzymatic activity [77].

Polysaccharide vaccines are derived from bacterial polysaccharide capsules and primarily induce B cell immune responses [78]. However, these vaccines are unable to elicit T cell immune responses as polysaccharides are considered T-independent antigens [79]. To enhance immunogenicity and promote long-term immunity, polysaccharide vaccines are often combined with immunogenic carrier proteins and form conjugate vaccines [80]. Currently, there are five licensed immunogenic carrier proteins: genetically modified cross-reacting material (CRM), tetanus (T) toxoid, diphtheria (D) toxoid, outer membrane protein complex (OMPC), and *Haemophilus influenzae* protein D (HiD) [81]. CRM is a toxin isolated from *Corynebacterium diphtheriae* and is often modified through mutations to reduce toxicity [82]. T and D toxoids are detoxified toxins obtained from *Clostridium tetani* and *Corynebacterium diphtheriae*, respectively, through formaldehyde treatment [83,84]. OMPC and HiD are non-toxic proteins derived from the outer membrane of *Neisseria meningitidis* serogroup B and *Haemophilus influenzae*, respectively [81,85].

VLPs are composed of self-assembling virus capsid proteins derived from various systems, such as bacteria, yeast, insect cells, plant cells, mammalian cells, and cell-free cultures. These extracted proteins can either self-assemble into single-protein VLPs, combine with other proteins to form multi-protein VLPs, or incorporate antigenic materials to enhance immune responses (chimeric VLPs). Since VLPs mimic the structure of pathogens and contain antigens themselves, they can stimulate immune responses, inducing both humoral and cell-mediated immunity, similar to natural infections [86,87].

The production of toxoid vaccines involves harvesting and purifying exotoxins, followed by their inactivation using heat, formaldehyde, or iodoacetamide [88,89]. Subsequently, protein subunits comprising major domains are used instead of full-length proteins to avoid potential immunostimulatory effects caused by activated CD4^+^ T cells and macrophages [90]. Toxoid vaccines are effective against pathogens that utilize toxins as their primary infection mechanism, as they stimulate the production of anti-toxoid antibodies that can neutralize pathogenic toxins [90].

Different subunit vaccines can be employed to effectively control infections depending on the type of pathogen. For instance, in the case of SARS-CoV and MERS-CoV, subunit vaccines have targeted the spike (S) proteins, which play a crucial role in infection by binding to host receptors [91,92,93]. As another example, the majority of research for influenza virus subunit vaccines has been performed with the HA protein, due to its high immunogenicity and conservability. As a result, several HA protein vaccines, including Fluvirin (1998), Agriflu (2009), and Flucelvax (2012), have been developed and FDA-approved [94]. Apart from HA proteins, another class of membrane proteins known as the extracellular domain of matrix protein 2 (M2e) has been explored for subunit vaccine development against influenza viruses [95]. M2e proteins are highly conserved and abundant in the cell membrane across all influenza A virus subtypes, providing cross-protective immunity when included in vaccines. However, M2e proteins have low immunogenicity and often require additional highly immunogenic carriers or adjuvants, posing challenges to the development of M2e vaccines beyond their current research stage [95]. In contrast, HA proteins are known to be more immunogenic than M2e proteins, making them an even more appealing candidate for subunit vaccines [96].

### 2.7. Adjuvants

Adjuvants are commonly used alongside vaccines to enhance their performance, including rapid response, long-lasting memory, and reduction in dose [97]. While whole-particle-based vaccines usually do not require adjuvants due to the presence of inherent adjuvant molecules, such as lipopolysaccharide, flagellin, and cytosine-phosphate-guanine, adjuvants are primarily used with subunit vaccines, as well as some DNA and mRNA vaccines [98]. Delivery vehicles with adjuvanticity have garnered attention as a promising type of adjuvant due to their versatility and ability to be combined with various molecules, such as other adjuvants, anchor proteins, and stabilizers. These delivery vehicles not only facilitate the transportation of the vaccine contents to targeted cells with controlled drug release but also enhance immune responses through a number of mechanisms.

Different types of delivery vehicles with adjuvanticity are currently under development, as shown in Table 1. Oil-in-water (O/W) emulsions are extensively studied adjuvants, characterized by a simple composition of tween and/or span, as well as a straightforward production process [99]. However, O/W emulsions have limitations, such as low thermal stability and relatively large particle sizes (<50–500 µm in diameter), which restrict their usage [100,101]. Conversely, inorganic micro/nanoparticles offer high thermal stability and can be flexibly shaped into many different forms. The drug release of inorganic micro/nanoparticles can be regulated by adjusting the amount of reactive moieties involved in degradation processes, such as hydrolysis (silica nanoparticles) and pH sensitivity (calcium phosphate nanoparticles) [102]. In comparison to other types of inorganic micro/nanoparticles, gold nanoparticles are known to be inert, enabling them to form stable functionalizations or have multi-layer coatings to enhance the bioavailability and immune responses of vaccines [103]. Nevertheless, long-term safety concerns persist with these vehicles due to their high stability [104].

In comparison to inorganic nanoparticles, protein-based nanoparticles (PNPs), commonly derived from animal and plant sources, have garnered attention due to their superior biodegradability, biocompatibility, and safety profiles. PNPs have demonstrated several other advantages, such as a convenient and affordable manufacturing process (attributed to the abundant presence of proteins in nature) and the possibility of modification for controlled delivery and specific targeting [150,151,152]. Some of the most commonly explored protein candidates for PNPs include human serum albumin, gelatin, and zein, while different PNPs exhibit different stability and immunogenicity profiles [123,153]. However, PNPs are also known for relative disadvantages, including challenges in controlling size and vaccine dose [153].

Lipid-based micro/nanoparticles are currently used as delivery vehicles for various types of commercial vaccines. These vehicles can encapsulate both hydrophilic and lipophilic particles due to their bilayer lipid structure [154]. They demonstrate high delivery efficiency, primarily because of their positive charge, which promotes their interaction with negatively charged antigen-presenting cells (APCs) [109]. However, it is important to note that this positive charge of lipid-based vehicles can also have side effects, such as destabilization of cell membranes by interacting with negatively charged proteins on the cell membrane [155].

Among the synthetic polymers used for vaccine delivery, poly(lactic-co-glycolic acid) (PLGA) and poly(lactic acid) (PLA) have found widespread application in vaccine delivery systems [156]. PLA exhibits a slower degradation rate compared to poly(glycolic acid) (PGA) due to the presence of a methyl group in its backbone, imparting hydrophobicity and reducing hydrolysis reactions [157]. Additionally, PLGA, the combined form of PLA and PGA, allows controlled drug release by adjusting the proportion of each polymer [158]. However, these synthetic polymers are limited by acidic byproducts, which can cause inflammation. In contrast, biopolymers are naturally occurring materials with relatively low toxicity. For example, acetalated dextran has been extensively used as a biopolymer in vaccine delivery systems due to its biocompatibility and the ease of modulating degradation time by adjusting the proportions of acyclic acetals (lower stability) and cyclic acetals (higher stability) to modulate degradation time [159].

### 2.8. Future Prospective

Whole-particle-based vaccines offer distinct advantages, such as strong immunogenicity and the ability to induce immune responses similar to natural infections. However, their production process is considered time-consuming and energy-intensive, making them cost-ineffective. Additionally, these vaccines can be immunostimulatory as they involve the injection of external materials. To address these challenges, gene- or protein-particle-based vaccines have emerged as alternative options. However, these vaccines often lack sufficient immunogenicity and transfection efficiency [86]. As a result, they are often encapsulated or coupled with self-adjuvating delivery vehicles. Nonetheless, these vehicles face limitations in terms of toxicity, stability, and low immunogenicity.

To overcome these limitations, new technologies have been developed, including various forms of nanomaterials such as hydrogel nanoparticles, carbon nanotubes, and dendrimers [86,160,161]. Moreover, various nanomaterials have demonstrated high efficiency in drug delivery, showing potential for inducing both humoral and cell-mediated immune responses and overcoming the limitations of conventional delivery vehicles [162,163,164,165]. For instance, hydrogel nanoparticles and carbon nanotubes can bind to bioactive molecules for long periods, resulting in sustained immunity [161]. As another example, cationic dendrimers, which form complexes with nucleic acids, do not destabilize cell membranes like lipid-based vehicles. Furthermore, due to their small sizes (1–10 nm), they can be encapsulated in other types of vehicles for additional functionalization [155].

To ensure the successful development of novel self-adjuvating carriers, several requirements need to be met, including stability, non-toxicity, biodegradability, immunogenicity, and controlled drug release. Commercializing these innovative technologies requires conducting pre-clinical and clinical research to evaluate their safety profiles. Overall, these emerging advancements in vaccine research are expected to bring about a positive transformation in vaccine development, offering safer and more effective treatment options for disease outbreaks.

## 3. Administration of Non-Invasive Vaccines

Non-invasive vaccines have gained significant recognition due to the drawbacks associated with conventional needle- and liquid-based vaccines, such as the need for cold chain storage, challenges of self-administration, issues with thermal and long-term stability, risks of needle-mediated infections, and needlestick injuries. Among the major non-invasive vaccine systems, oral, intranasal, and transcutaneous administration have been extensively investigated. These non-invasive approaches offer solutions to various limitations of invasive vaccines. However, the effectiveness of non-invasive vaccines is hindered by their limited bioavailability, as biological barriers prevent their efficient absorption into the body. To overcome this issue and enhance the bioavailability of non-invasive vaccines, diverse formulations tailored to specific administration routes have been developed.

### 3.1. Oral Administration

Oral vaccination has demonstrated its effectiveness in stimulating both mucosal and systemic immune responses by inducing the production of IgA and IgG antibodies [166]. The immune responses are mainly initiated in the intestine, where the majority of vaccine uptake and drug absorption occur [167,168]. The intestine consists of several layers, including the mucus layer, water layer, epithelial layer, basement membrane/Peyer’s patches, and lymph nodes [169]. The epithelial layer is composed of enterocytes, goblet cells, and microfold cells (M cells), which are joined together by tight junctions that prevent the movement of molecules through paracellular and transcellular routes [169,170,171]. Enterocytes, the most abundant cells in the intestinal epithelium (constituting up to 80% of the local cell population), transport antibodies through transcytosis using the neonatal Fc receptor (FcRn) and form antibody–antigen complexes [172,173]. Goblet cells produce mucin, the main component of the mucus layer, which provides physical protection against the uptake of pathogenic particles [174]. M cells play a crucial role in initiating immune responses by phagocytosing large particles and endocytosing small particles through specialized receptors like pattern recognition receptors (PRRs) [170,171]. Pathogen particles phagocytosed by M cells are subsequently taken up by dendritic cells (DC cells) in Peyer’s patches, leading to the activation of cellular and humoral immunity through the differentiation of T cells and B cells (Figure 4) [175].

Despite the advantageous characteristics of oral immunization, such as stability, independence from cold chain requirements, and ease of administration and storage, there are technical challenges in controlling the behavior of orally administered vaccines. Factors such as the varying pH levels of gastrointestinal fluids (e.g., stomach: 0.8–5, duodenum: ~7, jejunum: ≥7, ileum: ≥7, and colon: 7–8), osmolality, surface tension, viscosity, temperature, volume, hydrodynamics, composition, gastric emptying rate and force, intestinal transit time, and flow rate can impact the performance of orally administered vaccines [167,176]. Additionally, the presence of mucosal barriers and biomolecules, such as digestive enzymes, limits the bioavailability and immunogenicity of oral vaccines [77]. This low efficiency necessitates higher vaccine doses, which can eventually lead to immune tolerance in the host. Moreover, the primary approach for oral vaccines involves the use of heat-killed or attenuated pathogens, which makes it difficult to effectively activate the immune system due to poor penetration of the mucus layer. There is also a risk of attenuated pathogens regaining their toxicity, as exemplified by the oral polio vaccine (OPV) outbreak that occurred in several countries in 2000, including Egypt, Haiti, the Philippines, and the Dominican Republic. Extensive research has been conducted to address these limitations and optimize the administration of oral vaccines [177].

#### 3.1.1. pH Sensitivity

Enhancing the bioavailability of oral vaccines requires them to withstand the low-pH environment of the stomach. To improve vaccine stability, researchers have developed anionic coatings that remain stable under acidic conditions. These coatings consist of anionic polymers containing carboxylic acid groups, which remain unionized in the stomach’s low pH but become ionized in the higher pH of the intestine [178]. One example is Eudragit^®^, anionic polymers composed of poly(methacrylic acid-co-acrylates) with different chemical compositions [179]. The pH sensitivity of these polymers can be adjusted by varying the number of active carboxylic groups; higher levels of carboxylic groups result in increased sensitivity to low pH. For instance, Eudragit L100 (EL100) contains 48.3% active carboxylic groups and dissolves above pH 6, while Eudragit S100 (ES100), with 29.2% active groups, dissolves above pH 7 [179]. pH sensitivity can also be modified by using ethyl acrylate instead of methyl methacrylate in the polymer formulation, as seen with Eudragit L100-55 (EL100-55), which dissolves at lower pH (>pH 5.5) [180]. Cellulose-based materials, such as hydroxypropyl methylcellulose phthalate (HPMC-P), cellulose acetate phthalate (CAP), hydroxypropyl methylcellulose acetate succinate (HPMC-AS), and cellulose acetate trimellitate (CAT), have also been explored [181]. Among these, HPMC-P and CAP are most commonly used for enteric coatings and their pH-responsive behavior is based on the carboxylic acid residues that remain unionized in the stomach but become ionized in the intestine [182]. However, HPMC-P coating processes involving organic solvents raise concerns about toxicity and safety. Moreover, HPMC-P and CAP have limited thermal stability, restricting their application in capsule production [183]. Lastly, alginate, derived from brown algae, such as *Laminaria hyperborea*, *Laminaria digitata*, *Laminaria japonica*, *Ascophyllum nodosum*, and *Macrocystis pyrifera*, has been extensively used to produce coating polymers due to its biocompatibility, low toxicity, and affordability. Alginate’s anionic property also enables it to resist gastric fluid, making it a promising candidate for coating oral vaccines [184,185].

#### 3.1.2. Mucoadhesive Interaction

Increasing the bioavailability of oral vaccines requires enhancing mucoadhesion, which prolongs the retention time in the intestine [186,187,188,189]. To improve mucoadhesion, it is important to understand the behavior of different mucoadhesive materials. The mucoadhesive materials typically have hydrophilic functional groups (e.g., hydroxyl and carboxyl groups), a large molecular weight, high surface-to-volume ratio, and low cross-linking density to maximize interactions with intestinal mucus [190]. Among various natural and synthetic mucoadhesive enhancers, chitosan and carbomer have been widely studied [191].

Chitosan, a natural cationic polymer, interacts with negatively charged sialic groups in mucin through electrostatic attraction. Based on the mucoadhesive effects of chitosan, different chitosan-based conjugates (i.e., chitosan–cysteine, chitosan–glutathione, chitosan–thioglycolic acid, etc.) have been developed for vaccine delivery [187,189]. However, chitosan has limitations such as high solubility in the stomach. To address this issue, chitosan is often conjugated with alginate for coating purposes. The carboxyl groups of alginate interact with the amino groups of chitosan, resulting in strong electrostatic attraction. This alginate–chitosan complex shrinks and forms a gel in the low gastric pH, releasing drugs in the neutral pH of the intestine, enabling targeted drug delivery to the colon [192]. A similar study has also been performed with BSA to develop the alginate–chitosan complex as a vaccine delivery carrier, which has demonstrated the efficiency of the complex at a low pH [193].

Carbomer, a carboxyvinyl polymer containing large amounts of carboxylic groups (56–68%), forms hydrogen bonds with sialic acid and sulfate groups present on mucin glycoproteins’ oligosaccharide chains, thereby increasing mucoadhesive effects [194,195]. Carbopol^®^ 971P, 974P, and 934P are mucoadhesive polymers developed based on carbomer, and their mucoadhesive effects are further improved by combining them with cysteine, which interacts with mucus glycoproteins [196].

Nowadays, nanofibers have also gained attention due to their favorable properties such as high surface-to-volume ratio, high porosity, solubility enhancement, controlled cargo release, and the ability to target various sites (e.g., buccal, vaginal, oral, sublingual, transdermal, gastric, intestinal, colonic, and ocular mucosa). However, across different types of polymers, mucoadhesive effects have intrinsic limitations, including the relatively short adherence time (less than 4–5 h) for cells to absorb vaccines and the challenge of vaccine penetration through the mucus layer [197,198].

#### 3.1.3. Intestinal Permeability

To improve the bioavailability of oral vaccines, it is important to enhance their intestinal permeability due to the presence of the mucus layer that acts as a physical barrier [197,199]. This can be achieved through the use of small-sized particles, non-reactive coatings, or permeation enhancers (Figure 5). Nanoparticles have been extensively used for this purpose because of their ability to easily couple with functional groups and penetrate the mucus layer. It has been reported that anionic nanoparticles themselves have shown a relaxation effect on tight junctions, thereby enhancing the permeability, possibly due to interactions between the nanoparticles and integrin proteins on epithelial cells and the activation of myosin light chain kinase (MLCK) [200]. Furthermore, there have been studies demonstrating the size effect of particles on the penetration of the mucus layer, showing that large particles (1190 nm) experience a 30-fold reduction in particle penetration rate than small particles (510 nm) due to steric obstruction and capture by the mucus mesh (mucus mesh spacing ≥500 nm) [201,202].

Non-reactive nanoparticles can be created by functionalizing them with poly(ethylene glycol) (PEG) and Pluronic F127 to enhance their permeability. In particular, the development of low-molecular-weight PEG coating was inspired by the penetration of viruses through human mucus, reducing hydrophobic and electrostatic interactions between nanoparticles and the mucus layer through a hydrophilic and neutral-charged coating [203]. PEG has been successfully coupled with various types of nanoparticles (e.g., PLGA, poly sebacic acid (PSA), polyethylenimine (PEI), poly-l-lysine (PLL)) and has demonstrated efficient penetration [204]. Pluronic F127, a triblock copolymer surfactant with alternating hydrophilic and hydrophobic structures, exhibits a unique arrangement that minimizes its interaction with the mucus layer. Furthermore, its neutral charge allows it to avoid interactions with negatively charged mucins [205]. It has been reported that Pluronic F127-coated particles show an 80-fold increase in penetration rate compared to uncoated particles [186].

Chemical or enzymatic treatments, such as disulfide-breaking agents or mucolytic enzymes, have been also performed to enhance the intestinal permeability. Such permeation enhancers are typically incorporated inside or immobilized on nanoparticles to selectively affect certain regions of the mucus layer to maintain the functions of mucus layer against pathogens in non-treated areas [186]. N-acetyl-l-cysteine (NAC) is an example of a disulfide-breaking agent that breaks the disulfide bonds of cysteine groups in mucin using free sulfur groups [206,207]. Cleaving these bonds reduces the elasticity and viscosity of mucus, thereby increasing drug permeability through the epithelial layer [204]. Additionally, it is speculated that NAC may enhance the uptake of antigens by loosening the mucus layer at Peyer’s patches, where antigen uptake primarily occurs [174]. Other mucolytic enzymes, such as trypsin, papain, bromelain, and recombinant human DNase (rhDNase), can also contribute to reducing the viscosity of mucus by cleaving the cross-links in mucus glycoproteins [204,206].

### 3.2. Intranasal Administration

Intranasal vaccination offers several advantages, including easy accessibility, the potential for herd immunization, and the ability to initiate both mucosal and systemic immune responses, making it a favorable administration route for vaccines [208]. Similar to the Peyer’s patches in the intestinal tract, the nasal-associated lymphoid tissues (NALTs) play a significant role in inducing immunity through intranasal vaccination in rodents. NALTs are regions rich in immune cells (such as T cells, B cells, dendritic cells, macrophages, and M cells) where active immune responses occur [209]. The key difference between the Peyer’s patches and NALTs is that the formation of NALT is triggered after exposure to pathogens, whereas Peyer’s patches develop before birth. Similar to the NALTs of rodents, Waldeyer’s rings play a major role in human immune responses upon intranasal administration. Waldeyer’s rings, including the pharyngeal tonsil, tubal tonsils, palatine tonsils, and lingual tonsils, firstly transfer pathogens to DC cells, initiating mucosal immunity through the differentiation and proliferation of T cells [209,210]. Like oral vaccination, intranasal vaccination faces challenges from the mucus layer, which can result in mucociliary clearance of vaccine particles and hinder the vaccination process. Therefore, numerous methods have been investigated to enhance the immune response during intranasal vaccination, such as improving mucoadhesive interaction and intestinal permeability, as well as incorporating adjuvants.

#### 3.2.1. Liquid Formulation

Currently, intranasal vaccines are predominantly administered in liquid form due to their effective cell penetration. Liquid formulations allow for even distribution of vaccine components, such as adjuvants and stabilizers [211]. Developing a liquid formulation involves considering the properties of the liquid phase (e.g., osmolarity, pH, viscosity, and surface tension), the delivery devices, and the inclusion of stabilizers. In terms of osmolarity, isotonic solutions (~290 osmol/L) are commonly used to achieve optimal osmolarity, as hypotonic solutions are poorly absorbed and hypertonic solutions can cause cell shrinkage upon diffusion [211]. The pH of liquid formulations should be maintained between 4.5 and 6.5 to avoid irritation, considering that the nasal mucus layer has a pH of 5.5–6.5; it has also been observed that lysozymes in the intranasal system are inactivated at neutral pH, which can increase the risk of microbial infection [212,213]. Increasing the viscosity of nasal mucus is another approach to enhance the immunogenicity of vaccines by prolonging their retention time in the nasal cavity and increasing the dosing volume of the nasal spray over conventional dosing volume (50–140 µL) [214,215,216]. However, it must be noted that high mucosal viscosity may impede effective diffusion and spreading of the vaccines. Surface tension is another critical factor affecting the behavior of liquid vaccines. Most intranasal vaccines have lower surface tension (30.3–44.9 dynes/cm) than the nasal mucus layer (<56 dynes/cm). [217] The low surface tension of vaccines primarily facilitates better bioavailability by maximizing their spread within the nasal cavity [218].

To enhance the delivery efficiency of intranasal formulations, various devices have been developed, ranging from manual dropping-based devices to multifunctional devices with filters and automatic systems that prevent particle deposition in the lungs and ensure reproducibility [219]. Examples of devices used for liquid formulations include AccuSpray (FluMist), Classic Mexican Nebulizer (Measles), Aerovax and OptiMist (influenza), ViaNase (Alzheimer’s), and MAD and CPSI Cartridge Pump System (adenovirus-based vaccine) [209]. Among the different intranasal vaccination methods and devices, AccuSpray (FluMist) is the only one approved by the FDA for human use in the United States [220].

Due to the low stability of liquid formulations, stabilizers are often incorporated. For instance, FluMist contains monosodium glutamate, hydrolyzed porcine gelatin, and sucrose [221]. Other commonly used stabilizers in intranasal vaccines include lactose, sorbitol, porcine gelatin, arginine, and tricine [222]. These stabilizers can also be used during lyophilization to reduce the glass transition temperature and decrease the lyophilization time [223]. Moreover, sugars such as lactose and sucrose do not form sharp ice crystals, unlike water, providing physical protection to vaccines at low temperatures [224].

#### 3.2.2. Powder Formulation

Powder formulations can resolve limitations of liquid formulations, such as short residence time in the nasal cavity, low stability in terms of chemical and microbiological factors, short shelf-life, and the requirement for cold chain systems [209,225]. Dry powder vaccines are primarily manufactured using two methods: spray-drying and freeze-drying. Spray-drying starts with a liquid vaccine being sprayed into a drying chamber, followed by powder separation in a gas stream. On the other hand, freeze-drying involves freezing a liquid vaccine using liquid nitrogen and then subjecting it to lyophilization [226]. In terms of formulation process, dry powder vaccines share many similarities with conventional injectable vaccines, as well as antigens and adjuvants. However, one difference is that dry powder vaccines often incorporate bulking agents, stabilizers, and mucoadhesive materials to enhance nasal deposition [227]. Bulking agents play an important role as a vaccine carrier, facilitating the attainment of appropriate particle sizes to improve vaccine efficiency. For instance, lactose and mannitol, both FDA-approved inactive ingredients for nasal or respiratory use, are frequently chosen as carriers [228]. For a stabilizer of vaccines, mild formaldehyde has also been extensively used as it interacts with the residues of various antigens, such as the N-terminus of amino acids and side chains (i.e., arginine, cysteine, histidine, etc.) [227,229].

Dry powder vaccines are typically administered using medical devices like the μcoTM system and Opt-powder (influenza), Aktiv-Dry PuffHaler^®^ and BD solvent^®^ (measles), and Combitips Plus Syringe (meningococcal) [209]. While there are several advantageous aspects of powder formulations over liquid counterparts, limitations still exist due to limited research on powder formulation delivery systems and the behavior of dry powder vaccines upon contact with the nasal cavity [211,225]. This is evident considering that there are no FDA-approved dry powder vaccines currently available.

### 3.3. Transcutaneous Administration

Transcutaneous immunization (TCI) is a topical vaccination strategy that primarily stimulates systemic immunity, also inducing a moderate level of mucosal immunity with the assistance of an adjuvant [230]. TCI specifically targets epidermal cells, located in the middle layer of the skin, which also includes keratinocytes (KCs) and Langerhans cells (LCs). KCs are the predominant cells in the epidermis and play a role in recognizing pathogens through Toll-like receptors (TLRs), triggering innate and adaptive immune responses. LCs, a type of dendritic cells, capture a significant portion of TCI particles and recognize them through the major histocompatibility complex (MHC), activating adaptive immune responses [231]. However, for TCI to be effective, it is necessary to penetrate the stratum corneum, the outer layer of the skin consisting of 10–20 layers of dead skin cells that act as a barrier against external substances [232]. Consequently, research has focused on exploring physical and chemical methods to enhance the penetration of the stratum corneum in order to achieve successful TCI.

#### 3.3.1. Physical Delivery Systems

To enhance the penetration of vaccines through the stratum corneum, various methods have been employed, including ultrasound, electroporation, jet injectors, laser-based technologies, and microneedles [233]. Ultrasound can be utilized to create cavities in the stratum corneum, increasing vaccine permeability. This process involves the pressure gradient generated by ultrasound, enlarging air pockets in the fibrous tissue to allow the diffusion of vaccines through the cavities [234,235]. Electroporation, on the other hand, involves applying short high-voltage pulses to the cell membrane, temporarily damaging it and creating transient hydrophilic pores that enhance membrane permeability [236]. However, both ultrasound and electroporation require large machines and trained personnel to operate them. As an alternative, jet injectors, which are compact and easy to handle, have been used. These injectors use high-pressure liquid to deliver vaccines through the stratum corneum [237,238]. Despite the increasing accessibility, jet injectors have drawbacks including pain, bleeding, and edema [232].

Currently, diverse laser-based technologies (e.g., laser ablation, fractional laser technology, and ablative fractional laser technology) have also been developed to enhance stratum corneum penetration for vaccine delivery. Laser ablation is a conventional technology used to remove the entire surface of the skin, whereas fractional laser creates micro-columns in the skin, leaving the surrounding areas intact to minimize skin damage and accelerate the healing process [239]. Ablative fractional laser (AFL) is a method that combines the laser ablation and fractional laser technologies, minimizing skin damage and enhancing vaccine permeability by ablating the stratum corneum and forming skin microchannels [240].

Among the available transcutaneous immunization (TCI) methods, microneedles have gained acceptance due to their ease of use, accessibility, and independence from patches. Microneedles are minimally invasive and pain-free, involving small needles that penetrate the stratum corneum [241]. There are five categories of microneedles based on their structures: solid, coated, hollow, dissolving, and hydrogel-forming microneedles [242,243,244]. Solid microneedles are used to create microchannels in the skin, allowing for vaccine penetration [245]. To date, a variety of materials have been considered candidates for the production of solid microneedles (e.g., silicon, polysilicon, silicon dioxide, silicon nitride, PGA) [246]. While solid microneedles offer easy manufacturing and high dose delivery, they are hindered by a slower rate of diffusion when compared to other types of microneedles [247,248].

Coated microneedles contain vaccines applied to the needle, enabling diffusion into the epidermis upon injection. However, one limitation is that the coating may cause coated particle loss during manufacturing and handling [241,249]. As another option, hollow microneedles resemble conventional needle injection-based vaccination, with vaccines contained in the bore of the needle. These microneedles also provide the highest level of particle delivery but can experience blockage by tissues upon contact, although repositioning the bore to the side can address this issue [241,244].

Dissolving microneedles are made of water-soluble materials that penetrate and dissolve in the skin simultaneously. For these microneedles, vaccines and the microneedle matrix undergo a mixing process upon manufacturing. Among various candidates for the matrix of dissolving needles (e.g., sodium carboxymethylcellulose, poly(vinylalcohol), poly(vinylpyrrolidone), methylvinylether-co-maleic anhydride), sodium hyaluronate is most commonly used due to its biodegradability and mechanical strength [250,251]. Remarkably, dissolving microneedles have shown high diffusion efficiency within a patch wearing time of just several minutes, while it could be up to several hours for solid microneedles [247].

Lastly, hydrogel-forming microneedles have emerged as a promising option for their biocompatibility and controllable degradability [252]. These microneedles are equipped with hydrogel, capitalizing on its three-dimensional structure that allows for a significant capacity to carry vaccines; however, there is a possibility that vaccines can become trapped within the patch and they require a longer diffusion time of up to several hours [247,253,254]. The mechanism of hydrogel-forming microneedles relies on the absorption of interstitial skin fluid, which causes the hydrogel to swell and burst, triggering the release of the vaccines from the microneedle [254,255]. The vaccine-release behavior of the hydrogel is directly influenced by the swelling property of the matrix polymer and can be modulated by adjusting the amount of crosslinkers; increasing the percentage of crosslinkers results in a reduced swelling rate in the skin [252,254]. Last but not least, achieving optimal performance with hydrogel-forming microneedles requires maintaining their hardness in a dry state, while ensuring rapid swelling in the skin [255].

The efficiency of microneedle-based vaccines can be enhanced by using different types of nanoparticles with various strengths. For instance, polymeric nanoparticles offer controlled release and targeted delivery, while lipid-based nanoparticles provide enhanced solubility. Furthermore, inorganic nanoparticles are known to allow controlled release and have consistent pore structures, whereas protein-based nanoparticles enable controlled release and targeted delivery [256,257]. Currently, lipid-based carriers and polymeric carriers (i.e., poly(vinyl pyrrolidone) (PVP), Poly(vinyl alcohol) (PVA), PLGA, polypropylene sulfide, chitosan, etc.) have been the primary focus of research for vaccine delivery. These carriers have demonstrated enhanced immune responses without requiring the addition of adjuvants [242].

#### 3.3.2. Chemical Enhancers 

Various chemical substances, including water, dimethylsulphoxide (DMSO), acids, essential oils, and surfactants, have been investigated as potential enhancers for transcutaneous delivery [258]. These enhancers act to improve the permeability of vaccines through the skin by targeting different components of the stratum corneum: hydrophilic space (intercellular transport), hydrophilic lipid heads (transcellular transport), and hydrophobic lipid chains (transcellular transport) [259]. The hydrophilic space can accommodate solvents (e.g., propylene glycol, ethanol, pyrrolidines, dimethyl sulfoxide) through chemical treatment, allowing the delivery of vaccines and drugs in the stratum corneum, followed by their transport to the epidermis by a chemical gradient [259,260].

Water can interact with the hydrophilic heads of the bilipid membrane, increasing the fluidity and permeability of the stratum corneum, and this favorable property has led to the development of hydro patches used for vaccine penetration through the skin [232,261]. In a similar manner, DMSO ((CH_3_)_2_SO) can also effectively enhance the permeability of the stratum corneum; the oxygen atom present in the sulfoxide group of DMSO interacts with the hydrophilic heads of the bilipid membrane, causing a disruption of the hydrogen bonds between lipid heads. This process leads to a transformation of the gel-like structure in the ceramide bilayer of the stratum corneum into a liquid–crystalline structure [262]. On the other hand, lipophilic agents represented by azone, terpenes, and oleic acids disrupt the structure of lipid tails in the stratum corneum, expanding the membrane and facilitating vaccine diffusion [263]. However, these hydrophilic and lipophilic chemical enhancers have limitations in terms of efficiency and safety. Firstly, as the particle delivery into the epidermis is primarily mediated by chemical potential, the effectiveness of these agents diminishes as the concentration gradient decreases. This necessitates the application of mechanical force to enhance the efficiency of delivery. Furthermore, chemical molecules have the potential to affect tissue integrity and cause various forms of irritation, including local inflammation, erythema, swelling, dermatitis, and other related conditions [258,264].

### 3.4. Adjuvants

Currently, the majority of FDA-approved non-invasive vaccines utilize whole pathogen particles, such as Adenovirus Type 4 and Type 7 vaccines, VAXCHORA, ROTARIX, RotaTeq, Vivotif, and FluMist [220,265,266,267,268,269]. These vaccines typically do not require additional adjuvants, as the parts of the vaccines (e.g., lipopolysaccharide, flagellin, and cytosine-phosphate-guanine) have inherent adjuvanticity [98]. Regardless, there have been attempts to incorporate adjuvants into delivery systems to overcome the low bioavailability of non-invasive vaccinations (Table 2). Among the available adjuvants, aluminum-based adjuvants, including aluminum hydroxide and aluminum phosphate, have been the most widely used for over 80 years with invasive vaccines [270]. These adjuvants enhance immune responses by promoting antigen uptake by dendritic cells [271]. However, aluminum-based adjuvants have demonstrated lower immunogenicity for non-invasive vaccines compared to other adjuvants, likely due to their large particle size, which interferes with diffusion [272,273,274]. Consequently, efforts have been made to identify suitable adjuvants for non-invasive vaccines.

One alternative that has been studied is toxin-based adjuvants, such as cholera toxin (CT) and heat-labile enterotoxin (LT). However, their high toxicity raises concerns regarding their use. To mitigate this, mutations at specific amino acid sites of CT and LT have been introduced, resulting in improved efficiency and safety (e.g., double-mutant LT (dmLT) and multiple-mutated CT (mmCT)) [296]. These adjuvants are reported to specifically enhance the interleukin-17 (IL-17) response, thereby activating antigen-specific T helper cell responses [285,296].

In addition to toxin-based adjuvants, adjuvants based on other biological molecules (e.g., such as polysaccharides, lipids, proteins, and genes) have also been investigated. Polysaccharide-based adjuvants are primarily used to enhance the activity of immune cells, such as macrophages, and induce cytokine-mediated immune responses (e.g., IL-1 and -2 (mannatide), IL-5 and -6 (Advax™), IL-4 and -13 (chitin)) [297,298,299,300,301,302]. Nucleotide-based adjuvants (e.g., synthetic double-stranded RNA polyriboinosinic acid-polyribocytidylic acid [poly (I:C)], cytosine-phosphate-guanine oligodeoxynucleotide (CpG ODN), and cyclic guanosine monophosphate-adenosine monophosphate (2′3′-cGAMP or cGAMP)) are, on the other hand, target-specific molecules. For instance, poly (1:C), CpG ODN, and cGAMP specifically interact with Toll-like receptor (TLR) 3, TLR9, and stimulator of interferon genes (STING) receptors, respectively, activating inflammatory cytokines and type 1 interferon, eventually stimulating both innate and adaptive immune responses [301,302,303,304,305,306].

Lipid-based adjuvants (e.g., liposomes, lipid nanoparticles, emulsions, and immune-stimulating complexes (ISCOMs)) are often employed as vaccine carriers with adjuvanticity, capable of incorporating other additives, as well as facilitating interactions between vaccines and dendritic cells (DCs) [307]. Protein-based adjuvants (e.g., flagellin, poly(γ-glutamic acid), and proteosome), meanwhile, function as ligands for receptors (TLR5, TLR4, and TLR2, respectively) [308,309,310]. Their interactions activate helper T cells and cytotoxic T cells, thus stimulating adaptive immunity [311].

Despite the advantages offered by adjuvants, only a few have received FDA approval, including aluminum salts, adjuvant System 04 (AS04), oil-in-water emulsion, CpG 1018, and Quillaja saponaria 21 (QS-21) [312]. This delayed progress in developing vaccine adjuvants is primarily attributed to their reactogenicity and potential interactions with vaccines [313]. Consequently, in order to advance the development of vaccine adjuvants, extensive research into their interaction mechanisms with vaccines and the human body is essential, along with thorough preclinical and clinical studies aimed at establishing the safety profiles of emerging adjuvants.

### 3.5. Future Perspectives

Researchers have highlighted the numerous advantages of non-invasive vaccines compared to their invasive counterparts. One significant advantage is the potential to increase vaccination rates by eliminating needle-related fears and phobias that affect a substantial portion of children, adolescents (20–50%), and young adults (20–30%) [314]. Additionally, non-invasive vaccines are cost-effective by eliminating the need for cold chain systems, specialized facilities, and experienced personnel. Furthermore, disease-specific immunization can be achieved by developing and commercializing non-invasive vaccines. As shown in Table 2, Table 3 and Table 4, each administration method targets different diseases: intestine diseases (oral vaccines), respiratory diseases (intranasal vaccines), and outmost skin diseases (transcutaneous). Such an approach concentrates and maximizes immune responses at targeted sites. Additionally, in this sense, combination treatments of different non-invasive vaccines hold the potential for multi-site targeting. However, there is still a scarcity of approved non-invasive vaccines, as well as pre-clinical and clinical trials focused on non-invasive vaccines [24,29,77,315]. Therefore, further research endeavors should be directed towards addressing the challenges of non-invasive vaccines, optimizing their formulations and administration methods, in order to expand their usage and pave the way for effective global immunization against future disease outbreaks.

## 4. Conclusions

Despite the evident benefits of non-invasive vaccines, their commercialization has encountered physiological barriers that impede the efficient delivery of vaccines to immunological cells, such as antigen-presenting cells (APCs). Each non-invasive vaccine faces specific challenges that restrict their bioavailability. For example, oral administration is hindered by gastrointestinal fluids and mucosal layers, intranasal administration is hindered by mucosal layers, and transcutaneous administration is hindered by the stratum corneum. Therefore, to enhance the immunogenicity of non-invasive vaccines, the formulation of the vaccine should carefully consider limitations associated with each route of administration. Additionally, various supplementary materials can be employed in vaccine formulation to enhance bioavailability, such as pH-sensitive polymers, mucus-penetrating polymers, and microneedles for addressing gastrointestinal fluids, mucosal layers, and the stratum corneum, respectively.

To optimize the immunogenicity of vaccines and minimize their immune-toxicity, careful selection of vaccine types and adjuvants is also essential, as different combinations of these components can lead to varying immune responses and clinical outcomes. For instance, for non-invasive administration with low immunogenicity, adjuvants are often combined with whole-particle-based vaccines to boost their effectiveness. On the other hand, in the case of invasive administration, whole-particle-based vaccines are typically not paired with adjuvants due to their inherent high immunogenicity, whereas adjuvants are mostly used with subunit vaccines due to their lower immunogenicity.

Adjuvants serve not only as immune enhancers but also as carriers for vaccines. These carriers can be further functionalized with other types of adjuvants to bolster immune responses or coupled with anchor proteins and stabilizers for improved performance. Nevertheless, careful consideration must be given to the selection of adjuvants to be paired with vaccines, and their toxicity should be thoroughly assessed through pre-clinical and clinical tests prior to clinical application. Only with such precautions in practice will the development of effective, safe non-invasive vaccines be achieved.

## Figures and Tables

**Figure 1 pharmaceutics-15-02114-f001:**
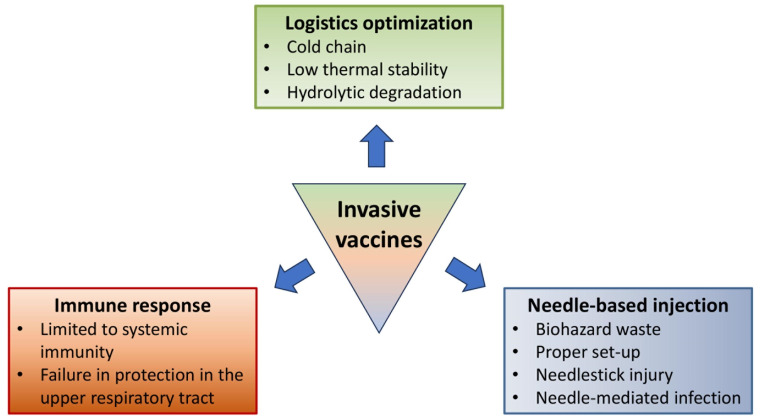
Schematic illustration of limitations of invasive vaccines.

**Figure 2 pharmaceutics-15-02114-f002:**
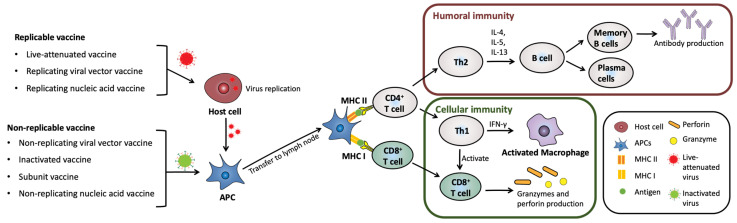
Schematic illustration of immune responses upon vaccine administration.

**Figure 3 pharmaceutics-15-02114-f003:**
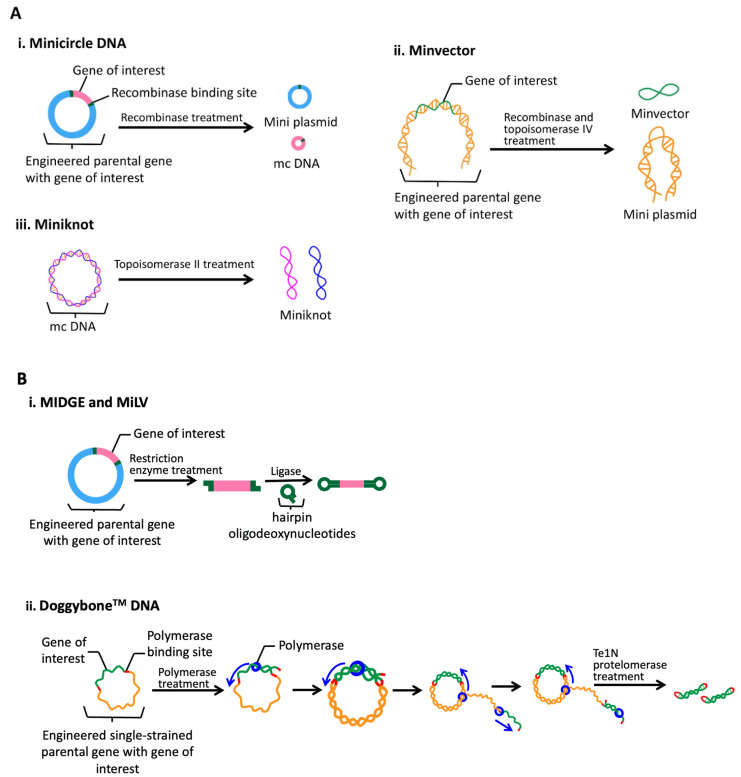
Schematic illustration of DNA vaccine production methods. Production of circular DNA (**A**) and linear DNA (**B**).

**Figure 4 pharmaceutics-15-02114-f004:**
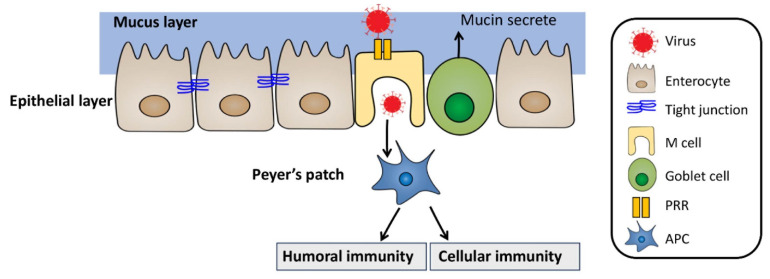
Representation of immune responses of an intestinal M cell.

**Figure 5 pharmaceutics-15-02114-f005:**
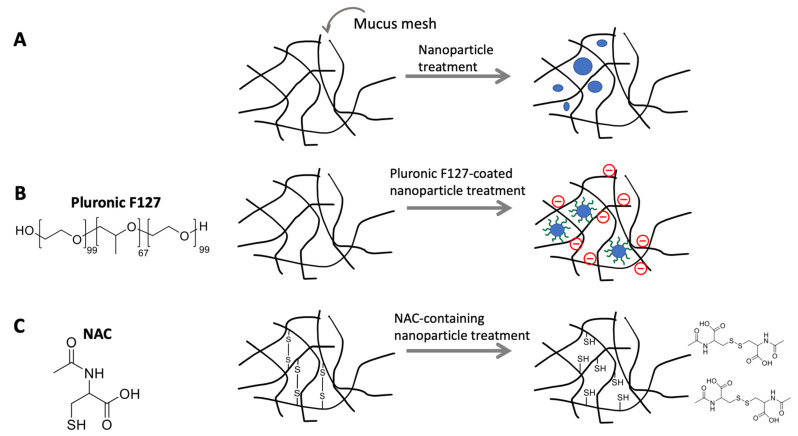
Schematic illustration of mucus-penetrating mechanisms. (**A**) Size-dependent penetration, (**B**) non-reactive coating treatment, and (**C**) permeation enhancer treatment.

**Table 1 pharmaceutics-15-02114-t001:** Nano- and micro-sized carriers with adjuvanticity.

Carriers/Adjuvants	Type	Immune Responses	Limitations	Status
**Lipid-based carriers**	Immune-stimulating complexes (ISCOM)	Activation of CD8 T cells [105] Th1 and Th2 activation [106]	High dose of saponin is required to manufacture (immunoreactive) [106] Limitation in encapsulation of non-membrane-derived antigens [106]	In clinical stage [107]
	Liposome	Th1 and Th2 activation [108]	Limitations in the selection of lipid due to rigidity [109]	In use in the vaccine industry; Pfizer/BioNTech and Moderna (SARS-CoV-2) [110]
	Bilosome	Th1 activation [111]	High stability but low immunogenicity [112]	In preclinical stage [113]
	Archaeosomes	Activation of CD8 T cells [114] Th1 and Th2 activation [115]	Mechanism-of-action- and safety-related research should be performed [115]	In preclinical stage [116]
	Virosomes	Activation of CD8 T cells [114,117] Th1 and Th2 activation [115,117]	Rapid disintegration in the blood [118]	In use in the vaccine industry; Epaxal^®^ and Inflexal^®^V (influenza vaccine) [119,120]
**Protein-based carriers**	Gelatin	Activation of CD8 T cells [121] Th1 and Th2 activation [122]	Large particle size Rapid degradation [123]	In clinical stage [124]
	Albumin	Activation of CD8 T cells [125] Th1 activation [126]	Potential immunogenic reaction [127]	FDA-approved [123]
	Zein	Th1 and Th2 activation [128]	Low stability [129]	In preclinical stage [128]
**Emulsion**	MF59 (oil-in-water emulsion)	Activation of CD8 T cells [130] Th2 activation [130]	Limitation in drug release control	In use in the vaccine industry; Fluad^®^, Focetria^®^, and Celtura^®^ (influenza vaccine) [130]
	AS03 (oil-in-water emulsion)	Th1 and Th2 activation [131]	Limitation in drug release control	In use in the vaccine industry; Pandemrix and Arepanrix (influenza vaccin) [132]
	AF03 (oil-in-water emulsion)	Th1 and Th2 activation [133]	Lower immune response compared to AS03 [133]	In use in the vaccine industry; Humenza™ (influenza vaccine) [134]
**Polymer**	Poly(lactic-co-glycolic acid)	Activation of CD8 T cells [135] Th1 and Th2 activation (Th2 dominant) [136]	Slow and non-tunable degradation rate [137] Degradation into acidic byproduct (inflammation) [138]	FDA-approved
	Poly(lactic acid)	Activation of CD8 T cells Th1 and Th2 activation [139]	Degradation into acidic byproduct (inflammation) [140]	FDA-approved
	Chitosan	Th1 and Th2 activation [141]	Reduction in CD8 T cell response [142]	FDA-approved
	Dextran (acetalated)	Activation of CD8 T cells [137] Th1 and Th2 activation [137]	Fast and tunable degradation rate [137] Degradation into pH-neutral byproduct [137,138]	FDA-approved
**Inorganic**	Calcium phosphate nanoparticles	Activation of CD8 T cells [102] Th1 and Th2 activation [143]	Low antigen loading capacity and rapid aggregation [102]	FDA-approved
	Silica nanoparticles	Activation of CD8 T cells [144] Th1 and Th2 activation [144]	Hemolysis through interaction of silanol groups and the phospholipids of the red blood cells [145]	In clinical stage [146]
	Gold nanoparticles	Activation of CD8 T cells [147] Th1 and Th2 activation [148]	Poor biodegradability [149]	FDA-approved

**Table 2 pharmaceutics-15-02114-t002:** Adjuvants for oral vaccines.

Adjuvant	Type	Vaccine Name	Vaccine Type	Target Disease	Status
**Toxin-based adjuvant**	Heat-labile enterotoxin (LT)	-	Inactivated vaccine [275]	*Helicobacter pylori*	In clinical
	Double-mutant heat-labile toxin (dmLT)	ETEC vaccine (ACE527) [276]	Live attenuated vaccine	Enterotoxigenic *Escherichia coli* (ETEC)	In clinical
		ETVAX [277]	Inactivated vaccine	Enterotoxigenic *Escherichia coli* (ETEC)	In clinical
		-	Inactivated vaccine [278] Subunit vaccine [279,280]	*Helicobacter pylori*	In preclinical
		-	Subunit vaccine [281]	*Clostridium difficile*	In preclinical
		-	Subunit vaccine [278]	*Clostridium tetani*	In preclinical
		-	Subunit vaccine [282]	Hepatitis B virus	In preclinical
		-	Live attenuated vaccine [283]	*Salmonella enteritidis*	In preclinical
	Multiple-mutated cholera toxin (mmCT)	-	Inactivated vaccine [284]	*Helicobacter pylori*	In preclinical
		-	Inactivated bacteria (Vibrio cholerae) [285] Subunit vaccine (influenza virus) [285]	*Vibrio cholerae* and influenza virus	In preclinical
	Recombinant cholera toxin B subunit (rCTB)	Dukoral [286]	Inactivated vaccine	*Vibrio cholerae*	Prequalified by WHO
		-	Live bacteria [287]	*Helicobacter pylori*	In preclinical
	Cholera-toxin-derived adjuvant (CTA1DD)	CTA1-3M2e-DD [288]	Subunit vaccine	Influenza virus	In preclinical
**Polysaccharide-based adjuvant**	Chitosan	-	Inactivated vaccine [289]	*Helicobacter pylori*	In preclinical
	β-glucans	-	Inactivated vaccine [290]	Influenza virus	In clinical
		-	Subunit vaccine [291]	*Salmonella Typhi*	In preclinical
		-	Inactivated vaccine [292]	*Bacillus anthracis*	In preclinical
	Arabinoxylan (AX)	-	Inactivated vaccine [290]	Influenza virus	In clinical
	Bacterial exopolysaccharide (EPS)	-	Inactivated vaccine [290]	Influenza virus	In clinical
**Lipid-based adjuvant**	α-galactosyl ceramide	-	Subunit vaccine [293]	Human immunodeficiency virus	In preclinical
		-	Inactivated vaccine [294]	*Vibrio cholerae*	In preclinical
		-	Inactivated vaccine [295]	*Helicobacter pylori*	In preclinical

**Table 3 pharmaceutics-15-02114-t003:** Adjuvants for intranasal vaccines.

Adjuvant	Type	Vaccine Name	Vaccine Type	Target Disease	Status
**Toxin-based adjuvant**	Heat-labile enterotoxin (LT)	Nasalflu (Berna Biotech) [316]	Inactivated vaccine	Influenza virus	In clinical
		-	Inactivated vaccine [317]	Influenza virus	In clinical
	Enzymatic A1 domain of LT (LTA1)	-	Subunit vaccine [318]	Influenza virus	In preclinical
	Cholera-toxin-derived adjuvant (CTA1DD)	-	Inactivated vaccine [319]	Influenza virus	In clinical
		-	Inactivated vaccine [320]	Human respiratory syncytial virus (hRSV)	In preclinical
		-	Subunit vaccine [321]	*Mycobacterium tuberculosis*	In preclinical
		-	Subunit vaccine [322,323]	Influenza virus	In preclinical
	Cholera toxin B subunit (CTB)	-	Live attenuated vaccine [323]	Influenza virus	In preclinical
		-	Subunit vaccine [324]	Influenza virus	In preclinical
**Polysaccharide-based adjuvant**	RS09 (Ala-Pro-Pro-His-Ala-Leu-Ser)	-	Viral vector vaccine [325]	Human immunodeficiency virus	In preclinical
	Mannatide (polyactin A)	-	Inactivated vaccine [326]	Influenza virus	In preclinical
	Advax™	-	Live attenuated [327]	Influenza virus	In preclinical
	Chitosan	-	Non-viral vector vaccine [328]	*Enterohemorrhagic Escherichia coli* (EHEC) O157:H7	In preclinical
		-	Inactivated vaccine [329]	Influenza virus	In preclinical
		-	Subunit vaccine [330,331]	Influenza virus	In preclinical
		-	Subunit vaccine [332]	Hepatitis B virus	In preclinical
	Chitin	-	Live attenuated [323]	Influenza virus	In preclinical
		-	Subunit vaccine [333]	Influenza virus	In preclinical
**Nucleotide-based adjuvant, synthetic adjuvant**	Synthetic double-stranded RNA polyriboinosinic acid-polyribocytidylic acid [poly (I:C)]	-	Live attenuated vaccine [323]	Influenza virus	In preclinical
		-	Inactivated vaccine [334]	Influenza virus	In preclinical
	Cytosine-phosphate-guanine oligodeoxynucleotide (CpG ODN)	-	Subunit vaccine [335]	Influenza virus	In preclinical
		-	Inactivated vaccine [336]	Enterovirus	In preclinical
		-	Subunit vaccine [337]	SARS-CoV-2	In preclinical
		-	Non-viral vector vaccine [338]	*Pseudomonas aeruginosa*	In preclinical
		-	Non-viral vector vaccine [339]	*Escherichia coli*	In preclinical
	Cyclic guanosine monophosphate-adenosine monophosphate (2′3′-cGAMP or cGAMP)	-	Subunit vaccine [340]	SARS-CoV-2	In preclinical
**Aluminum-based adjuvant**	Alhydrogel^®^ (aluminum oxyhydroxide gel)	-	Subunit vaccine [341]	SARS-CoV-2	In preclinical
**Lipid-based adjuvant**	Endocine™ (lipids monoolein and oleic acid)	-	Subunit vaccine [342]	Influenza virus	In preclinical
**Protein-based adjuvant**	Flagellin	-	Inactivated vaccine [343]	SARS-CoV-2	In preclinical
		-	Inactivated vaccine [344]	Influenza virus	In preclinical
	Poly(γ-glutamic acid)	-	Inactivated vaccine [334]	Influenza virus	In preclinical
	Proteosome	-	Subunit vaccine [273]	SARS-CoV-2	In preclinical
		-	Inactivated vaccine [345]	Influenza virus	In preclinical
	Protollin^TM^	-	Subunit vaccine [310]	Influenza virus	In preclinical

**Table 4 pharmaceutics-15-02114-t004:** Adjuvants for transcutaneous vaccines.

Adjuvant	Type	Vaccine Name	Vaccine Type	Target Disease	Administration Method	Status
**Toxin derivates**	Double-mutant heat-labile toxin (dmLT)	-	Subunit vaccine [346]	Nontypeable Haemophilus influenzae (NTHI)	Hydration (sterile, pyrogen-free 0.9% sodium chloride)	In preclinical
	Single-mutant heat-labile enterotoxin	-	Subunit vaccine [347]	Enterotoxigenic Escherichia coli (ETEC)	Hydration and abrasion (70% isopropyl alcohol, glycerol 10%, and 10 swipes of medical grade sandpaper)	In clinical
		-	Subunit vaccine [348]	*Corynebacterium diphtheriae*	Hydration (sterile PBS)	In preclinical
		-	Subunit vaccine [349]	*Clostridium tetani*	Hydration	In preclinical
	Cholera toxin (CT)	-	Subunit vaccine [272]	*Corynebacterium diphtheriae*	Hollow microneedle (pre-treatment purpose)	In preclinical
		-	Inactivated vaccine [350]	Influenza virus	Hydration (saline-soaked gauze)	In preclinical
	Cholera toxin B subunit (CTB)	-	Subunit vaccine [351]	hepatitis B virus	Solid microneedle and hydrogel patch	In preclinical
**Polysaccharide-based adjuvant**	Chitosan	-	Subunit vaccine [352]	*Corynebacterium diphtheriae*	Hollow microneedle (pre- or post-treatment purpose)	In preclinical
**Nucleotide-based adjuvant, synthetic adjuvant**	Synthetic double-stranded RNA polyriboinosinic acid-polyribocytidylic acid [poly (I:C)]	-	Subunit vaccine [353]	Influenza virus	Coated microneedle	In preclinical
	Cytosine-phosphate-guanine oligodeoxynucleotide (CpG ODN)	-	Subunit vaccine [354]	*Clostridium tetani*	Hydration (PBS-drenched wound plaster)	In preclinical
		-	Subunit vaccine [272]	*Corynebacterium diphtheriae*	Hollow microneedle (pre-treatment purpose)	In preclinical
		-	Non-viral vector vaccine [355]	*Chlamydia muridarum*	Hydration (sterile PBS)	In preclinical
		-	Subunit vaccine [356]	Human immunodeficiency virus	Hydration (saline-drenched gauze)	In preclinical
		-	Subunit vaccine [357]	Foot-and-mouth disease virus	Hydration	In preclinical
**Immune-stimulating complexes (ISCOMs)**	Quil A	-	Subunit vaccine [272]	*Corynebacterium diphtheriae*	Hollow microneedle (pre-treatment purpose)	In preclinical

## Data Availability

Not applicable.

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
