# Peer review of "Non-Invasive Vaccines: Challenges in Formulation and Vaccine Adjuvants"

_pharmaceutics, 2023, doi:10.3390/pharmaceutics15082114_

Round 1
Reviewer 1 Report
This review introduces non-invasive vaccines and the challenges in formulation and vaccine adjuvants. The overall feeling after reading is the lack of focus in introducing the topic though the review is long and contains lots of information. There are 4 major sections: introduction (1), types of vaccines and their formulation (2), administration of non-invasive vaccines (3), conclusions (4). The definition of invasive and non-invasive vaccines is in the first session. The characterization of needle-based injection vaccines as invasive vaccines is somewhat overstated considering the majority of current vaccines are delivered by needle injection and the overall good safety of needle-injected vaccines. To avoid confusion, the tone or the word ‘invasive’ may need to be changed. In the second session, the replicable DNA and RNA vaccines were not introduced. Virus-like particle (VLP) vaccines can be introduced in the subunit vaccine subsession considering FDA already approved HPV and HBV VLP-based vaccines. The majority of seasonal influenza vaccines are based on purified or expressed HA antigens, which can be introduced before the M2e-based experimental vaccines under development. It’s recommended to clearly state the vaccine platforms proved vs. under development to allow readers to catch the current status of the field. It’s recommended to follow traditional characterization method to divide adjuvants into different types in Table 1 (combined system and lipid are confusing and not clearly differentiated from each other). In the third session, the major introduction is obtained from drug delivery, which should be tailored for vaccine delivery (the word ‘drug’ showed up in many places). Examples should be given related to vaccine delivery when introducing the methods to overcome barriers for oral, intranasal, or transcutaneous delivery. For transcutaneous delivery, laser technologies can also be mentioned. Different adjuvants were listed in Table 2-4 for different routes of delivery. The underlying reasons of a specific adjuvant for a particular route of delivery can be introduced. Or else, there is no need to divide adjuvants based on the route of administration. In Table 3, gene-based adjuvant is more accurately described as ‘nucleotide-based adjuvant’. In Table 4, LPS should be removed since it’s less likely to advance to clinical studies due to the safety concern. Conclusion can be expanded to a few paragraphs. Line 248, mRNA transcription is often carried out in vitro nowadays in preparation of mRNA vaccines.
Overall the English is easy to understand.
Reviewer 2 Report
This review is a well-written and organized one that deals with the non-invasive methods for vaccination.
A lot of effort was exerted so I recommend its acceptance after performing the following:
The use of protein nanoparticles such as gelatin, albumin, zein NPs in vaccine delivery should be added to the review and Table 1 so that the review becomes complete.
Also the use of microneedles or needle arrays in delivering protein NPs containing adjuvants or model adjuvants should be added to the microneedles part in the non-invasive methods.
Acceptable
Reviewer 3 Report
Authors reviewed “non-invasive vaccines” in general. The review is well organized and the topic are interesting although they are broad. The manuscript is useful for general readers. Hopefully, the size of the manuscript should be more compact.
Minor comments
Line 140. “2” in “NH-CH2OH” should be subscript.
Lines 445. 452, 454. “M-cells” reads “M cells”. (see line 588, “M cells”)
Line 472. “inactivated pathogens” reads “attenuated pathogens”?
Line 757. “2” in “((CH3)2SO” should be subscript.
Tables: The formats of tables can be improved. Reconsider them.
Round 2
Reviewer 1 Report
Authors addressed most of the concerns in the revised manuscript. Regarding dividing vaccines into invasive and non-invasive vaccines, I still think it’s not accurate and easy to cause confusion since no invasive vaccines will be approved. You can use invasive, non-invasive, or minimally invasive to describe vaccine delivery methods. Vaccines are either safe or non-safe. Another statement needs to be changed. Line 243-245 mentioned mRNA vaccines expressing TLRs and cited ref 58. The overall statement is not right since mRNA vaccines don’t express TLRs except introducing the open reading frames of the TLR genes. Ref. 58 is about the stimulation of host cells to express TLRs due to Covid-19 infection. It’s recommended authors carefully read the manuscript and correct any wrong statement.
